# Exploring the challenges and opportunities of multisectoral nutrition programme in Ethiopia: A qualitative study on combating undernutrition during pregnancy

Nana Chea[1]*, Judith den Hertog[2], Marlon Lukken[2], Julie van der Houven[2], Milou Dirks[2], Ayalew Astatkie[1], Mark Spigt[2,]

1 School of Public Health, College of Medicine and Health Sciences, Hawassa University, Hawassa, Ethiopia, 2 Department of Family Medicine, CAPHRI School for Public Health and Primary Care, Maastricht University, Maastricht, the Netherlands

* cheanana2007@gmail.com

## Abstract

### Background

Despite evidence of the benefits of multisectoral nutrition programmes in reducing undernutrition among vulnerable groups such as pregnant women, implementation is not as expected for reasons that are not well understood. This study aimed to explore the challenges, opportunities, and strategies of a multisectoral nutrition programme in improving undernutrition among pregnant women in Ethiopia.

### Methods

A phenomenological qualitative study was conducted in four rural districts of Sidama Region, Ethiopia, in 2023. Thirty-six in-depth interviews and four focus group discussions were conducted with programme coordinators, sector office leaders, and nutrition specialists from governmental and non-governmental organizations. Data were audio-recorded, transcribed, translated, and thematic analysis was carried out using qualitative data analysis software, MAXQDA version 2022.

### Results

The study identified key challenges and opportunities for a multisectoral nutrition programme aimed at addressing undernutrition among vulnerable groups such as pregnant women. Challenges include a lack of operational structure, poor integration of sectoral activities, inadequate budgeting, and low commitment from collaborators and staff. However, opportunities exist in the form of existing government structures, positive attention from global organizations, its principles and existance of community centers. Optimal strategies to strengthen the programme include improved

**Data availability statement:** All relevant data are within the paper and its Supporting Information files.

**Funding:** The author(s) received no specific funding for this work.

**Competing interests:** The authors have declared that no competing interest exist.

**Abbreviations:** ANC, Antenatl Care; CSA, Central Static Agency; FGDs, Focus Group Discussions; IDIs, In-Depth Interview; IRB, Institutional Review Board; MAXQDA, MAX Qualitative Data Analysis; NGO, Non-Government Organization; NNP, National Nutrition Programme; PSI, Population Service International; PSNP, Productive Safety Net Programme; SVN, Stichting Vluchteling Nederland; UN, United Nations; UNSCN, United Nations Standing Committee for the Nutrition.

coordination, integration, communication, innovative financing, advocacy, capacity building, and community engagement.

## Conclusion

Despite many challenges, the multisectoral nutrition programme is a promising start in addressing undernutrition in general and among pregnant women in particular, using existing government structures with greater community and partner involvement and integration.

## Introduction

Undernutrition is a major global public health problem, disproportionately affecting vulnerable groups such as children and pregnant women in low-income countries [1]. In many low- and middle-income countries, including Ethiopia, multiple factors contribute to the high prevalence of undernutrition among pregnant women. Lack of information about optimal nutrition due to limited access to nutrition education, inaccurate or incomplete nutrition information is one of the reasons for the problem [2]. Culture and traditions that prohibit certain food groups may contribute to undernutrition among pregnant women. For example, a study conducted in southern Ethiopia found that about 27% of participants avoided food because of food taboos [3]. Similarly, in many parts of the world, gender inequality limits women's and girls' access to education, income and resources, including food. This leads to inadequate nutrition, especially for pregnant and lactating women, who have increased nutritional needs [4–7].

The Ethiopian government has initiated several policies to combat undernutrition. One of these was the historic Seqota Declaration, a multisectoral collaboration to combat stunting and wasting among children and pregnant women in the northern part of the country. Its primary goal was to eradicate childhood stunting and wasting by 2030. The programme was launched in 2015 and is currently in its second phase (expansion phase) on scaling up across the country. It aims to end the cycle of poverty by supporting healthy growth and nutrition in the first 1,000 days of life, with a lasting impact on subsequent generations; and in line with the United Nations Sustainable Development Goals [8]. In addition, the National Nutrition Programme (NNP) has been especially designed to address the eating habits of infants and young children, enhance maternal nutrition, and guard against micronutrient deficiencies. This programme includes both nutrition-specific and nutrition-sensitive interventions against undernutrition [9]. Furthermore, the country has been implementing the Productive Safety Net Programme (PSNP), a large, national, social safety net programme aimed at improving food security and reducing poverty among vulnerable populations [10].

Many studies have highlighted the importance of multisectoral collaboration in addressing nationwide health problems, such as undernutrition. According to an exploratory evaluation of 26 national nutrition plans, scaling up multisectoral

nutrition interventions is more important than ever to sustain gains made in the fight against all forms of undernutrition and to achieve the sustainable development goals [11]. Similarly, a review of nutrition programmes in developing countries found that those that used a multisectoral approach were more likely to be successful than those that relied solely on food supplementation [12]. A study in Bangladesh found that a multisectoral programme that combined nutrition education, agriculture, and health interventions was effective in reducing undernutrition among pregnant women [13].

Despite the evidence supporting multisectoral nutrition programmes to address undernutrition, many challenges remain in implementing such programmes. A study of a multisectoral programme in Latin America and Africa focusing on child undernutrition, found that poor coordination was one of the many challenges of this multisectoral programme [14]. Another study conducted in sub-Saharan Africa, identified lack of political commitment, limited resources, and poor coordination among sectors as major barriers to effective implementation of multisectoral cooperation [15]. The Ethiopia study also found that ambiguity in the roles of national and regional sectors, lack of awareness of the programme plan, and budget constraints were some of the challenges faced by the multisectoral nutrition programme. However, the source of this evidence was a comparable study in Ethiopia and Nepal, which did not report exclusively on the challenges of the programme in Ethiopia [16,17].

Therefore, the present study aimed to explore the challenges, opportunities and strategies of a multisectoral nutrition programme to improve undernutrition among pregnant women in Ethiopia.

## Methods and materials

### Study design

We conducted a qualitative phenomenological study to understand the challenges, opportunities, and strategies of multisectoral nutrition programme in addressing undernutrition, focusing on undernutrition among pregnant women in districts of Sidama National Regional State, Ethiopia, from May to June 2023.

### Study setting

We conducted this study in four districts of Sidama National Regional State, namely Loka Abaya, Boricha, Bilate Zuria, and Hawassa Zuria which are located in rural Ethiopia.

In 2019, Ethiopia's Central Statistical Agency (CSA) projected that pregnant women made up 3.46% of the region's population, while non-pregnant women made up 19.84%, live births 3.46%, infants under 1 year 3.19%, toddlers 5.18%, children under 2 years 5.18%, children under 5 years 15.5%, and people over 45 years 48.3%.

### Study participants and selection procedures

Out of the 30 districts in Sidama National Regional State, four districts, namely Loka Abaya, Boricha, Bilate Zuria, and Hawassa Zuria, were purposefully selected for this study. These are four of the twelve districts where a multisectoral nutrition programme is actively implemented. In addition, partners or non-governmental organizations (NGOs) namely Save the Children, Population Service International (PSI), Stichting Vluchteling Nederland (SVN), and the Seqota Declaration coordination office, were also included in the study. From each selected district, sector offices such as health, agriculture, water and minerals, and women, youth and children were included in the study because of their active technical involvement in the programme. Finally, we selected one from each sector office head, multisectoral programme coordinator, and nutrition coordinator and NGO involved in the present study. Their expertise, position and knowledge relevant to the topic of interest were reasons for selecting participants. Similarly, from the NGOs, programme or nutrition coordinators were involved in the study. They were included in the study considering their active work involvement in nutrition-related activities and multisectorial nutrition programme in the region.

 

## Sample size

We ended up with thirty-six in-depth interviews (IDIs) in four districts. Specifically, sixteen IDIs were conducted with sector office heads, twelve IDIs with nutrition coordinators, four IDIs with multisector nutrition programme coordinators and four additional IDIs with the programme coordinators of NGOs were conducted.

Regarding the focus group discussions (FGDs), we had conducted four FGDs with 6–8 participants per FGD. Two FGDs with sector office heads, one FGD with nutrition and programme coordinators, and the last one with NGO programme coordinators. We decided to discontinue further FGDs due to redundancy of information and lack of new insights.

## Data collection process

Health professionals with extensive experience in qualitative research data collection, including the primary and co-authors, facilitated and collected data using face-to-face IDI and FGD techniques in *Sidaamu Afoo* (the local language). Some of the co-authors conducted both analytical and reflective memos throughout the data collection process. They used an interview topic guide to facilitate possible probes. They conducted IDIs and FGDs at the workplace of the interviewees and discussants, in a quiet place on the premises of selected sectors. All interviews and discussions were audio recorded and supplemented with notes taken during the interveiew and discussion.

## Data management and analysis

Audio recordings were transcribed and then translated from the local language into English. The English transcripts were uploaded into MAXQDA version 2022, a qualitative data analysis software, for efficient management and analysis. The process began with coding for elements of interest within the data and then organizing these codes. The renowned six-phase guide by Braun and Clark, offering a practical structure for conducting such analysis, was referenced for this project [18]. To achieve objectivity in the coding process, two researchers openly coded the generated transcripts using an inductive approach. They also checked the consistency of the coding on a daily basis during peer debriefing. In case of disagreement, a third researcher was consulted to reach a common understanding of the coding. The two researchers then reviewed the revised set of codes for textual congruence and linkage using axial coding. Next, similar codes were systematically categorized and categories were labeled. Finally, the labeled categories were transformed into candidate themes and final non-repetitive themes. Two of the authors assessed the main themes and sub-themes in relation to the transcripts and interview guides. In general, a hybrid approach of thematic analysis, which combines both deductive and inductive approach, was employed, and the results were cross-checked for consistency. The deductive analysis involved using the semi-structured interview guide as a framework to categorize themes and responses, while the inductive analysis uncovered themes that naturally arose from the transcripts., and the final themes were decided upon to be described in the report.

## Quality and reflective statement

Prior to data collection, the authors discussed and refined the topic guide (a semi-structured questionnaire) and had it translated into the local language, *Sidaamu Afoo*. To ensure its effectiveness and validity, a pre-test was conducted in a comparable district (Shebedino District). During the data collection phase, trained data collectors collected data under the daily supervision of the primary author.

The researchers were aware of their positionality and its potential influence on the study. With different educational backgrounds, including a PhD in public health and a Master of Science in community health, they understood how their individual experiences and worldviews could influence the procedures used to collect and analyze data. To address this, they actively practiced reflexivity, keeping a notebook in which they recorded their ideas, biases and and assumptions. They were able to critically examine their responsibilities and how their positions might have influnced the interpretation of

results. During a preview validation workshop, they also asked participants for feedback from their identities through membership checks. The researchers sought to reduce bias and improve the rigour and credibility of their qualitative study by adopting reflexivity. Their commitment to analyze their own opinions significantly increased overall strength of the study.

**Ethics approval and consent to participate.** Before starting the data collection, we received ethical approval from the Institutional Review Board (IRB) of the College of Medicine and Health Sciences at Hawassa University. The Sidama Regional Health Bureau, district (woreda) administrations, and selected facilities were consulted in writing before proceeding. Before the interview or focus group discussion began, a written informed consent was secured from all study participants.

**Inclusivity in global research.** Additional information regarding the ethical, cultural, and scientific considerations specific to inclusivity in global research is included in the Supporting Information.

## Results

### Socio-demographic characteristics of the study participants

All planned participants were included in the study. They expressed an interest in taking part during the introductory discussion for consent and showed their commitment during the interviews. The participants in this study were aged between 28 and 45 years. They came from a variety of professional backgrounds, including public health officers, managers, water engineers, agricultural engineers, and nutritionists (Table 1).

### Challenges, opportunities and optimal strategies of the multisectoral nutrition programme

From the analysis, challenges, opportunities and strategies emerged as major themes, and many sub-themes also emerged under each major theme. Common sub-themes related to challenges included lack of operational structure, poor integration, inadequate budgeting and limited staff commitment and awareness. Similarly, sub-themes related to opportunities were the existing government structure, positive attention from the international organization, the principles of multisectoral nutrition programme and community centers. Sub-themes related to strategies included strengthening coordination and integration, innovative financing, resource mobilization, advocacy, capacity building and community engagement.

### Challenges

Under this main theme, four sub-themes were identified. Each of the sub-themes has been narrated with supporting quotations as follows.

**Lack of well-defined programme structure.** Participants discussed a major challenge of the multisectoral nutrition programme in addressing malnutrition in general and among pregnant women in particular. Participants identified the lack of a working structure for the multisectoral nutrition programme as the main obstacle to its effective implementation. According to the participants, the main obstacles were structural, such as the lack of a designated office and staff, and the lack of uniform standards of procedures for working together across sectors. Each sector had worked to its own timetable and convenience, resulting in a lack of common goals and objectives. In the districts, programme implementers focused on children rather than pregnant women, even though children and pregnant and lactating women are the targeted beneficiaries of the programme. Participants felt that the programme lacked written and unwritten instructions on how to deal with transport issues, especially in remote rural areas, which added to the challenge of effectively implementing and monitoring a multi-sectoral nutrition programme to address undernutrition among pregnant women.

*"…I mean each sector is providing service to the community to improve the hungry status of the community, especially undernutrition among children but not pregnant women. However, all offices such as health, agriculture, water, and gender are not working in structured way." [FGD participant #7, 40 years old]*

**Table 1. Socio-demographic characteristics of the participants involved in the focus group discussions and interviews.**

| Characteristics | | Frequency of participants | Percentage (%) |
|---|---|---|---|
| A: Socio-demographic characteristics of the focus group discussants May 2023 | | | |
| Age category (in years) | <18-24 | 9 | 30 |
| | 25-34 | 11 | 37 |
| | ≥4035−49 | 8 | 27 |
| | ≥ 50 | 2 | 6 |
| Sex | Male | 19 | 63 |
| | Female | 11 | 37 |
| Marital status | Single | 8 | 27 |
| | Married | 23 | 77 |
| Occupation/profession | Public health | 6 | 20 |
| | Agricultural engineer | 2 | 7 |
| | Water engineer | 4 | 13 |
| | Management | 6 | 20 |
| | Nutrition | 4 | 13 |
| | Sociology | 5 | 17 |
| | Gender expert | 3 | 10 |
| Educational level | Diploma | 5 | 17 |
| | BSc | 12 | 40 |
| | MSc/MPH | 13 | 43 |
| Position | Sector leader | 16 | 53 |
| | programme coordinator | 10 | 33 |
| | Nutrition coordinator | 4 | 13 |
| Work Experience | < 5 years | 6 | 20 |
| | 6-10 years | 11 | 37 |
| | 11-15 years | 9 | 30 |
| | >15 years | 4 | 13 |
| B: Socio-demographic characteristics of the in-depth interview participants May 2023 | | | |
| Age category (in years) | <18-24 | 10 | 28 |
| | 25-34 | 15 | 42 |
| | 35-49 | 7 | 19 |
| | ≥50 | 4 | 11 |
| Sex | Male | 27 | 75 |
| | Female | 9 | 25 |
| Marital status | Single | 6 | 17 |
| | Married | 30 | 83 |
| Occupation/profession | Public health | 5 | 14 |
| | Rural agriculture professional | 6 | 17 |
| | Water engineer | 5 | 14 |
| | Management | 6 | 17 |
| | Nutrition | 4 | 11 |
| | Sociology | 4 | 11 |
| | Gender expert | 6 | 17 |
| Educational level | Diploma | 10 | 28 |
| | BSc | 17 | 47 |
| | MSc/MPH | 9 | 25 |
| Position | Sector leader | 16 | 45 |

*(Continued)*

**Table 1.** (Continued)

| Characteristics | | Frequency of participants | Percentage (%) |
|---|---|---|---|
| | Programme coordinators | 4 | 11 |
| | Nutrition coordinators | 4 | 11 |
| | NGO programme coordinators | 4 | 11 |
| | Technical assistant | 8 | 22 |
| Work Experience | < 5 years | 8 | 22 |
| | 6-10 years | 10 | 28 |
| | 11-15 years | 11 | 31 |
| | >15 years | 7 | 19 |

Abbreviations: BSc, Bachelor of Science; MSc, Master of Science; MPH, Master of Public Health

*"From 'seqota', we distributed some goats and hens to poor households, but that is to feed milk and eggs to their children, not to feed pregnant women."[IDI participant, 30 years old]*

Study participants also reported that the programme has not structurally included all partners, working to address nutrition problem of the community.

*"… we have been invited to attend the meeting of the multisectoral nutrition programme sometimes but structurally we are not the member of the regional multisectoral nutrition programme."[IDI participant, 39 years old]*

**Poor integration of sectoral activities of the multisectoral nutrition programme.** The multisectoral nutrition programme was designed to provide comprehensive and holistic solutions to complex problems, particularly undernutrition among vulnerable groups such as children and pregnant women. However, lack of integration has resulted in fragmented service delivery, with different sectors providing disjointed services towards the same goal. This created confusion, overlap, or gaps in services in combating undernutrition in the community in general and in combating undernutrition among pregnant women. Open and transparent communication channels are lacking, as well as willingness to share knowledge and expertise across sectors. A thematic analysis of the data from the various sectors showed that, in contrast to the multisectoral nutrition programme's goals, each sector was pursuing separate goals and employing different approaches. For example, data from the women, youth, and children affair sector office showed that they are working to change the behavior of pregnant women to avoid food taboos. Similarly, data from the health office showed that they have a health education and communication programme to change the nutrition related behavior of the pregnant women. In addition, the analysis report from the water and mining office reveals that this office is working to improve the water access problem of the community in general and of the vulnerable groups like children and pregnant women in particular. However, ideas raised from the study participants showed that sectors were not working together.

*"… Our office works day and night for the community to get water continuously. Regarding collaboration with the health office, our office provides what they need when request comes from their side…..I don't think that we have been working together." [IDI participant, 40 years old]*

Poor integration is also exacerbated by speculated power imbalances and mistrust between sectors. Sectors with greater resources or influence dominated decision-making processes, marginalizing the contributions of other sectors. Accordingly, resources are being allocated differently to similar activities by the sectors. This showed inefficient use of limited resources and could result in competing initiatives that undermined each other's effectiveness.

*"Most of the time, sectors office other than health office consider multisectoral nutrition programme is health sector's office programme. Off course health office is the secretay of this programme and they have more budget to run this programme activities."[IDI participant, 32 years old].*

The analysis of data from NGOs showed that they were working to empower women and children. However, their discussions revealed that their activities were not integrated with those of the government sector in addressing the nutritional problems of the community in general and of pregnant women in particular.

*"… Activity overlap with government office is a problem…."* [IDI participant, 45 years old)

**Lack of programme budget and logistics.** Almost all participants in the study mentioned the grossly inadequate funding and logistical support for the programme. They stressed that addressing complex nutrition issues requires significant financial commitment and resources. The programme encompasses a range of interventions, from the provision of selected animal and plant seedlings to the establishment of community nutrition demonstration laboratories, the organisation of gender-sensitive forums and the procurement of raw materials. Each of these critical components requires sufficient funding to be effectively implemented and sustained. However, participants reported that the programme has suffered from a critical shortage of funds, which has hindered the successful implementation of its activities. This lack of resources undermines the programme's potential to have a meaningful impact on the complex nutrition challenges it seeks to address, particularly in reaching remote pregnant women.

*"The budget for the multisectoral nutrition programme is very small, I can say it is null. In the case of our agricultural office, there are 38 activities expected to be achieved, but the budget does not enable us to perform all of these activities. For example, in order to buy a tin of seeds for vegetables, the price had increased in comparison from the past. Despite this, the budget allocated to the sector was reduced. For example, the budget that was allocated for Bilate Zuria district agricultural office is 328,000 birr, which is a very small budget to accomplish 38 activities. When you see the transportation costs in order to see one kebele, which costs 200 birr, it is expensive…."* [FGD participant #, 43 years old]

The lack of dedicated funding for the programme posed significant challenges. It affected the ability to reach targeted groups such as pregnant women, and the fund was insufficient to cover transport costs, purchase necessary materials and facilitate effective communication, ultimately threatening the sustainability of the programme.

*"…the challenges we are facing includes budget shortage, transportation cost, decreased motivation due to lack of incentives, and price inflation of supplies which caused training programmes to be cancelled. Unable to achieve planned activities. For example, we planned to provide 4 chickens to each household. We have 30 households with pregnant undernourished women but we reached only 12 households …."* [IDI participant, 32 years old]

The inadequate programme budget resulted in poor knowledge and skills transfer among staff and the community. Due to lack of budget, it was not possible to conduct training and skills transfer workshops for staff and the community.

*"…due to lack of the budget, we are unable to conduct supportive supervision, training and workshops. We have nothing to buy stationery, to pay to the participants for the transportation and so on."[FGD participant #8, 35 years old]*

Many participants reported the shortage of logistics and transportation challenges to be associated with the shortage of the budget of the programme.

*"Due to shortage of budget, we were not able to make supportive supervision at health centers and health posts. We pay 200 birr for the motor bicycle for one trip, but we don't have incentive for going to supervise wether the household practise proper handling of the livestock provided to them."[IDI participant, 34 years old]*

It also emerged that the shortage of budget resulted from poor integration of the available budgets for nutrition.

*"…eeh our office has a budget for nutrition for pregnant and for lactating women as well as for the undernourished children. Save the Children also have budget. Similarly, government office also has budget. But if we bring it togetherr, we can improve budget shortage for nutrition."[IDI participant, 32 years old]*

**Poor commitment.** A significant number of study participants sindicated that the level of commitment and involvement of the people involved was below expectations. For example, although the regional president and district heads were formally appointed to lead the multisectoral nutrition programme in their respective regions and districts, they rarely participated in the programme. In the region, the chair of the programme is a person delegated by the president's office rather than the president himself. similarly, the sector programme technical coordinators have failed to prioritize the programme's goal, focusing instead on the routine activities of their sectors. This neglect could be attributed, in part, to the lack of additional incentives associated with the programme. In addition, there were reports of significant delays in the financial sector's approval and disbursement of the allocated budget, which further hindered the programme's progress.

*"Higher officials from the steering and technical committees were conspicuously absent. They are prioritizing their regular activities. Other political concerns receive more of their attention than this one. For example, for all the meetings this year, the chairman has sent his delegates, who is unaware of the goals and objectives of programme, instead of attending the meeting in person." [group IDI participant, 35 years old]*

In the study area, the nutrition coordinators of the multisectoral nutrition programme in each sector also showed a lack of commitment, which could be attributed to several factors. First, they perceived the programme as an additional burden on their already heavy workloads without incentives or recognition. This sense of being overburdened without reward could lead to negligence and a lack of motivation to actively participate in the programme. It takes a lot of commitment and motivation for staff to travel to serve pregnant women in remote areas, which was not the case.

*"programme staff are critical to moving initiatives forward. yet, despite their expertise and responsibilities they face motivation and compensation challenges.. They want recognition for their extra efforts."[group IDI participant, 45 years old]*

Second, there was a perception among some staff that the programme was not a government initiative, but rather an NGO efforts. This perception may have influenced their expectations of the programme.

*"Our office staff often perceive me as wealthy because of my role in coordinating a multisectoral programme. They associate the Seqota office with financial prosperity, noting that it is a local NGO known for its abundant resources and frequent training and workshop opportunities."[IDI participant, 27 years old]*

Moreover, participants stated that sectors offices involved in the programme were not adhering to their common goal rather than giving due attention to their respective sector office goals.

*"… pregnant women healthy diet feeding is not our office responsibility, health department should work for this case… kkkk, we can teach about gender equality." [FGD participant #4, 26 years old]*

**Poor programme awareness.** There seemed to be a general problem with awareness of the programme among coordinators, staff, and community members. Despite efforts to address malnutrition among vulnerable groups such as pregnant women, there is a lack of understanding of the importance of a coordinated response involving multiple sectors. Coordinators and staff may not fully understand the impact of their specific role within the larger context of the initiative. This can lead to a disconnect between the goals of the programme and the actions of those responsible for its implementation. In addition, community members who are key stakeholders may not be aware of the existence or purpose of the initiative, which can hinder their engagement and active participation in the fight against malnutrition among pregnant women and children.

*"Regarding the challenges, when we give some vital garden seeds to the pregnant women's households, the community members consider us as aid staff. Even our staff, coordinators and community members have poor awareness about the programme, some technical assistants complain for extra payment saying it is NGO's programme."[IDI participant, Unknown age]*

## Opportunities

**Existing structure of the government.** Many participants noted that existing government structures, such as the health and agricultural extension programmemes and the one-to-five community networks, especially the pregnant women's networks called the Women Health Development Army, provide a valuable opportunity for seamless implementation of the multisectoral nutrition programme. Within this framework, dedicated staff, such as health and agricultural extension workers, serve as crucial links to the community, facilitating collaboration and outreach.

Similarly, the existence of other nutrition programmemes such as a NNP, PSNP and others is another opportunity in this regard. However, these existing structures are currently underutilized and represent an untapped potential for the success of the programme. By using the network of health extension workers in the health sector and agricultural extension workers in the agricultural sector, the programme can effectively reach and engage communities. These established networks can serve as a solid foundation for coordinating efforts and ensuring the programme's impact at the grassroots level. It is important to recognize the value of these existing structures and explore ways to optimize their use to effectively achieve the programme's objectives.

*"… For example, when we want to show how a pregnant woman should cook and use diverse foods, we use HEWs. Similarly, when we distribute the productive seedling to the pregnant woman's household, agricultural extension workers have helped us a lot…." [FGD participant #5, 29 years old]*

**Global organization's attention towards the programme and its principle.** The interest of different international organization was reported by almost all participants in the discussions. First, the global organizations have increasingly recognized the importance of a multisectoral approach to nutrition. Global organizations such as the as the World Health Organization (WHO), the United Nations through its SDG of achieving zero hunger, food and agricultural organizations, and others are giving due attention to addressing nutrition problems through a multisectoral approach. Donors and development partners have prioritized multisectoral nutrition programmes in their funding portfolios. evidences indicated that children and pregnant women's undernutrition pooled the attention of the government and non-government organizations to fight it using a multisectoral nutrition programme.

*"… You know, there was a presentation by the Deputy Prime minister of Ethiopia and the MOH of Ethiopia where they were at a conference together to share and present the information on 'Seqota'declaration, and Manager of the World*

*Bank said that we invested So many billions on infrastructure, and there was no change. Hereafter, if we invest in nutrition for human development, we will bring about a change through multisectoral nutrition approach.We will invest in six African countries. As you know, in the Amhara region, the World Bank started supporting the 'Seqota' Declaration..."[IDI participant, 45 years old]*

The second issue mentioned by all participants as an important opportunity to leverage was the principles of the multisectoral nutrition programme. The principles underlying the multisectoral nutrition programme provide a solid framework for collaboration. They include steering committees, technical committees, demonstration labs, and civil and private sector involvement are conducive to implementing collaboration.

*"If they implement this programme as per what is stated in the programme document, I can say it is a wonderful start and philosophy and I hope it will lead to making undernutrition a story."[IDI participant, 35 years old]*

**Community centers.** Various community centers, such as churches, mosques, "edir" associations and "equb" groups have emerged as key sites for addressing nutrition problems, particularly opportunity to improve awareness including the dangers of undernutrition during pregnancy through community engagement and collective action. The people involved in these community centers can be sensitized to use the media readily available to them as a tactical tool to disseminate knowledge, raise awareness, and promote healthy eating practices throughout the community.

The perspectives shared by participants highlight the importance of a multimodal approach using both modern media platforms and physical community meeting places. Combining these channels increases the likelihood of reaching more people and ensures that important nutrition information is shared with everyone in the community. This inclusive approach not only increases the impact of the message but also gives community members the confidence to take charge of their own health and well-being.

Significant change can be achieved by harnessing the potential of community centers and utilizing the power of group action. By working together, community members can solve the nutritional problem faced by pregnant women and create a community that is healthier and more resilient for future generations.

*".. Uhh.. For example, if we could use existing community meeting places such as churches and 'edir' that have malnourished people, it would be easy to disseminate the the information about healthy eating to pregnant women and the community at large…." [FGD participant #4, 33 years old]*

**Optimal strategies**

**Strengthening coordination, integration and communication.** To address the challenges, participants suggested establishing clear coordination mechanisms and improving communication between sectors. This includes the development of standardized protocols, regular intersectorial meetings, and joint planning sessions. They also emphasized the need for a well-defined framework outlining the roles and responsibilities of each sector, including partners, and effective communication channels to facilitate collaboration. Strengthening the role of nutrition coordinators was also suggested to improve cohesion and ensure consistent messages across sectors.

*"… Uhhh I think the problem is not only the shortage of budget, but also we are not bringing nutrition funds together to fight undernutrition." [IDI participant, 45 years old]*

Participants also suggested that the programme should have a clear target group for implementation: Pregnant women, especially those living in remote rural areas, should be prioritized as targeted beneficiaries of the programme.

*"… I mean, although programme targeted on the undernourished pregnant women, children and lactating women, we did not do much for the pregnant women who is lives in far rural areas. There were too many women who had nothing to eat." [FGD participant #5, 29 years old]*

**Innovative financing and resource mobilization.** In response to budget constraints, participants proposed several innovative financing strategies. These included advocating for increased government funding for nutrition programmes, engaging the private sector through corporate social responsibility initiatives, and exploring international donor funding opportunities.

*"… this programme has no budget allocation from the government side. Ehhh,..why not? Staff should work on it. The community should also contribute…."[FGD participant #5, 44 years old]*

*"The government and donors need to be pushed to give more money… more attention is needed." [IDI participant, Unknown age]*

They also emphasized the importance of efficient use of resource and suggested the exploring cost-effective interventions such as community-based nutrition education and local food production initiatives. Diversifying funding sources and ensuring sustainable financing were highlighted as key components of a successful multisectoral nutrition programme. In addition, the study participants stressed that there should be a pooling of the scarce budget available to nutrition from different sectors, including NGOs.

*"If the multisectoral nutrition is to be sustainable and effective, the budget problem should be solved by bringing together all the budgets allocated for nutrition by government and NGO. I mean, there has not been an efficient use of the resources to combat undernutrition." [FGD participant #1, 34 years old]*

**Advocacy, capacity building, and community engagement.** To address challenges related to sector commitment and community involvement, participants recommended advocacy campaigns targeting sector leaders and community members. Capacity building initiatives, such as trainings and workshops, were suggested to raise awareness about maternal nutrition and engage key stakeholders. In addition, community engagement strategies, including involving husbands and community leaders in nutrition programmmeming, were suggested to increase community ownership and participation. Participants emphasized the need for culturally sensitive approaches and the use of community health workers to increase the reach and effectiveness of nutrition interventions.

*"…I recommend the Nutrition Get Together Forum which brings all partners and sectors working on nutrition in the region to improve the fragmented movement against undernutrition." [FGD participant #9, 31 years old]*

Participants also suggested that working to improve malnutrition in pregnant women would indirectly improve child malnutrition. That is, well-nourished pregnant women are likely to have a normal-weight baby.

*"From my involvement in various awareness programmes, I have learned that most community members, including some staff, have not yet understood that feeding pregnant women is indirectly feeding the unborn baby. I know that all organs, including the brain, are formed during pregnancy. So I think our programme should focus on pregnant women." [IDI participant, 39 years old]*

## Discussion

This study explored the challenges, opportunities, and optimal strategies for improving the implementation of a multisectoral nutrition programme in the case of undernutrition among pregnant women in the Sidama region Ethiopia.

The current study identified structural barriers as the main challenges facing the multisectoral nutrition programme in the study area. There is no clearly defined structure for integrating budget and activities and for working with partners such as the community and other organizations. Health Systems researcher Ruel noted in his study that "a key challenge in multisectoral nutrition programme is to define clear roles and responsibilities, and ensure effective coordination and integration of sectoral activities" [19]. Previous studies conducted in different parts of the world have reported similar findings. For example, a similar study in Ethiopia reported that work structure was a challenge for the multisectoral nutrition programme [17]. Other studies conducted in northern Burkina Faso and in seven World Bank-supported countries found that poor integration of programme sector activities can lead to inefficient use of resources, missed opportunities for holistic outcomes and impact, and weakened accountability and coordination [20,21]. Similarly, a report on scaling up nutrition in 26 countries found that integration of nutrition interventions is the backbone of successful multisectoral nutrition programmeming [11]. In support of the current study's findings, previous studies of multisectoral experiences in seven World Bank-supported countries, including Rwanda, the Democratic Republic of the Congo, Cambodia, Guatemala, Indonesia, Malawi, and Nigeria, reported that budget constraints were a common challenge for multisectoral nutrition programmes [21]. Another study, conducted by the Ethiopian NNP also concluded that lack of funding was the critical issue, facing multisectoral nutrition programmes [9]. Similarly, studies conducted in Pakistan, Mozambique and Nepal found that staffcommitment and awareness of the programme were low [21–23]. Studies from northwestern, northern, and southern parts of Ethiopia also reported structural and budgetary challenges to multisectoral nutrition programmes [23,24]. The results of the current study indicate that role confusion, duplication, lack of accountability, missed opportunities, poor coordination, low impact and sustainability are the result of lack of a well-defined programme structure. Therefore, designing a robust working structure [25], strengthening government commitment and prioritization of nutrition, promoting partnerships with private and improving community engagement can mitigate the funding challenge of the programme. In addition, building capacity through mentorship and training and strengthening supervision can create a sense of ownership and task prioritization among programme stakeholders and staff.

In terms of the opportunities for programme, this study identified the presence of an enabling environment as an untapped opportunity for implementing a multisectoral nutrition programme in the area. Favorable conditions and factors such as government structures, international attention, community resources, and cultural assets were untapped opportunities for the success of this programme. A synthesis of experiences, and lessons learned from Global Forum for Rural Advisory Services and the World Bank's Secure Nutrition Knowledge Platform collaboration concluded that the presence of extension workers is an opportunity to empower vulnerable individuals to make healthier food choices and improve their eating behaviors [26]. A Ugandan study found that the involvement of local government structures in nutrition planning was very important in addressing community nutrition problems [27]. Consistent with our current study, the United Nations Standing Committee on Nutrition [28] highlighted the failures of individual sectors in addressing malnutrition and their advocacy for the multisectoral alliance to effectively address the issue [28]. In addition, the WHO, in its publication"Systems Thinking for Health Systems Strengthening" also mentioned a systems approach to malnutrition as supportive [29]. Similarly, a study conducted in sub-Saharan Africa found that specific socio-cultural characteristics of each community need to be taken into account before implementing nutrition and health promotion programmes [15], suggesting that some cultures in society are supportive of nutrition programmes, while others may not be. Evidence from other literature has also shown that community centers have emerged as a promising way to implement multisectoral nutrition programmes [30]. These findings highlight that multisectoral nutrition programmes are a globally prioritized intervention modality whose impact and sustainability can be maximized by recognizing the unique socio-cultural characteristics of each community and the local structures available [31]. In addition, this study suggests that linking health and agricultural extension workers and community centers to the programme implementation strategy can improve programme success.

The study also explored the optimal strategy for improving the implementation of a multisectoral nutrition programme: comprehensive programme strengthening. This included strengthening coordination, integration and communication; innovative financing and resource mobilization; advocacy; capacity building; and community engagement. In support of this finding, Nigeria's National Health Sector Nutrition Strategic Plan and the Multisectoral Nutrition Strategy 2020–2025 both recommended similar approaches for the programme [32,33]. Several studies have also shown that innovative financing and resource mobilization are increasingly recognized as essential strategies for improving the multisectoral nutrition programeming [8,27,34]. A USAID study highlighted the importance of focusing on civil society capacity building for multisectoral nutrition advocacy [35]. A systematic review also showed the importance of community involvement for successful health programme initiation, process and implementation [36]. The results of the current study have several implications. First, diversification of funding sources may lead to stable and sustainable funding for this programme. Second, bringing together different partners, including the community, can alleviate the financial constraints of the multisectoral nutrition programme. In general, this strategy has the potential to provide more sustainable funding for the programme. Accordingly, mobilizing the community and partners to raise funds for the multisectoral nutrition programme should be part of the programme.

Interms of limitations and strengths of the study, the involvement of multidisciplinary researchers, the participation of officials from governmental and NGO, and the involvement of sector leaders or decision makers as study participants can be mentioned as the strengths of this study. This multidisciplinary approach enhances the rigor and validity of the research findings [37]. It also promotes innovative thinking and the integration of knowledge from different fields, which contributes to more effective solutions and policy recommendations [38]. In addition, Involving decision-makers in the study ensures that the study's findings and recommendations are relevant and actionable in the policy context [39]. However, the exclusion of federal leaders and programme end users from the study may be a limitation of this study. High-level leaders often have a broader perspective and influence on policies that affect multiple sectors and regions. By not including their perspectives, the result may miss important cross-sectoral dynamics and potential policy levers at higher levels of government [40]. Similarly, according to human-centered research design, involving people directly affected by the programme under study can provide important insights into the study [41].

## Conclusion

This study shows the challenges and opportunities of implementing a multisectoral nutrition programme in rural districts of the Sidama region of Ethiopia. The focus of these findings was on pregnant women, one of the programme's vulnerable populations. By highlighting the ill-defined programme structure, poor integration, and budget constraints, the findings suggest the importance of coordination, integration, and innovative financing. The existence of government structures and community networks provides unique opportunities to improve programme effectiveness. Recommended strategies, offered by experienced practitioners and decision makers, provide a roadmap to success, emphasizing improved collaboration and sustainability. By addressing challenges and capitalizing on opportunities, the region can strengthen its multisectoral nutrition programme and ultimately improve the nutritional outcomes and well-being of its community in general and pregnant women in particular.

## Supporting information

**S1 File.**
(DOCX)

**S1 Data.**
(PDF)

**S2 Data.**
(PDF)

**S3 Data.**
(PDF)

**S4 Data.**
(PDF)

**S5 Data.**
(PDF)

**S6 Data.**
(PDF)

## Acknowledgments

We are grateful to the study participants and the data collectors for their invaluable time, dedication, and overall support. We would like to express our gratitude to the Hawassa University for ethical clearance and the Sidama Regional Health Bureau for providing us a letter of official permission that was helpful during the data collection procedure.

## Author contributions

**Conceptualization:** Nana Hankalo, Judith den Hertog, Marlon Lukken, Julie Van der Houven, Milou Dirks, Ayalew Astatkie.

**Formal analysis:** Nana Hankalo, Judith den Hertog, Marlon Lukken, Julie Van der Houven, Milou Dirks, Ayalew Astatkie, Mark Spigt.

**Investigation:** Nana Hankalo, Marlon Lukken, Mark Spigt.

**Methodology:** Nana Hankalo, Judith den Hertog, Marlon Lukken, Julie Van der Houven, Milou Dirks, Mark Spigt.

**Software:** Judith den Hertog, Ayalew Astatkie.

**Supervision:** Nana Hankalo, Marlon Lukken, Julie Van der Houven, Milou Dirks, Ayalew Astatkie, Mark Spigt.

**Validation:** Marlon Lukken, Mark Spigt.

**Visualization:** Mark Spigt.

**Writing – original draft:** Nana Hankalo, Judith den Hertog, Marlon Lukken, Julie Van der Houven, Milou Dirks, Ayalew Astatkie, Mark Spigt.

**Writing – review & editing:** Nana Hankalo, Judith den Hertog, Marlon Lukken, Julie Van der Houven, Milou Dirks, Ayalew Astatkie, Mark Spigt.

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
