## [Decision Letter · Decision Letter 0]

Dear Dr. Hankalo,

Thank you for submitting your manuscript to PLOS ONE. After careful consideration, we feel that it has merit but does not fully meet PLOS ONE’s publication criteria as it currently stands. Therefore, we invite you to submit a revised version of the manuscript that addresses the points raised during the review process.

We look forward to receiving your revised manuscript.

Kind regards,

Dinaol Abdissa Fufa, Mph

Academic Editor

PLOS ONE

3. In the online submission form, you indicated that “The data that support the findings of this study are available on request from the corresponding author via cheanana2007@gmail.com ”

Reviewers' comments:

Reviewer's Responses to Questions

**Comments to the Author**

1. Is the manuscript technically sound, and do the data support the conclusions?

Reviewer #1: Yes

Reviewer #2: Yes

Reviewer #3: Yes

2. Has the statistical analysis been performed appropriately and rigorously?

Reviewer #1: Yes

Reviewer #2: No

Reviewer #3: N/A

3. Have the authors made all data underlying the findings in their manuscript fully available?

Reviewer #1: No

Reviewer #2: Yes

Reviewer #3: Yes

4. Is the manuscript presented in an intelligible fashion and written in standard English?

Reviewer #1: Yes

Reviewer #2: Yes

Reviewer #3: No

Reviewer #1: I congratulate the authors on the study and thank them for the opportunity to evaluate it.

In the introduction, I suggest showing more of the originality of this work: What is different? Why publish it?

I suggest improving the tables, presenting them in a more didactic and scientific way. How was this age division made?

Reviewer #2: The paper is technically sound but has several issues.

1. The authors highlight challenges seen in other countries on multisectoral nutrition programs but there was no background information about Ethiopia. How did you identify that this was a challenge in Ethiopia?

2. Line 100 - 108: What is the relevance of this information?

3. Line 125: I had to try and figure what what these stand for. Spell out IDIs and FGDs in the first instance for everyone to follow

4. Line 127: You say you had a preset sample size. So how did you use data saturation principle when you had already estimated 20? And why did you end up with 36 (line 128) when you had estimated 20?

5. Line 133: Why did you end up with four and not ten?

6. Line 135: If you were considering this, then the preset sample size does not work. What is the minimum for a qualitative study? That is what you use if you are using the data saturation principle.

7. Line 147: Write in the correct order of doing things

8. Line 151: It is still unclear which theoretical framework was used for this study. On what are you basing your work?

9. Line 153 - 157: How was objectivity achieved during the coding process? How many authors conducted coding and how were these compared?

10. The paper has several grammatical and spelling errors. I urge the authors to thoroughly read through and correct these.

Reviewer #3: This article examines a qualitative study to understand facilitators, barriers, and implications that address undernutrition in pregnant women participating in a multisectoral nutrition programme in Ethiopia. The authors provide sound evidence of a careful and considerate community-based study that utilized interviews and focus groups. However, there were many grammatical errors that made it hard to read the manuscript. I would suggest a major revision for editing. Some of the themes that were presented appear to have overlap and redundancy, specifically 'Lack of operational structure,' and 'Shortage of Programme Budget and Logistics.' Overall, well done study.

**Do you want your identity to be public for this peer review?** For information about this choice, including consent withdrawal, please see our Privacy Policy

Reviewer #1: No

Reviewer #2: No

Reviewer #3: No

---

## [Author Response · Author response to Decision Letter 1]

8 Apr 2025

On behalf of authors of the manuscript entitled “Exploring the Challenges and Opportunities of Multisectoral Nutrition Programme in Ethiopia: A Qualitative Study on Combating Undernutrition during Pregnancy”, I would like to pass our sincere thanks and appreciation to the reviewers and the editor of PLOS ONE Journal for giving us invaluable comments and feedback on our manuscript.

Appended with this letter are subsequent pages that hold point-by-point responses to comments forwarded by the reviewers and Editorial Board. The revisions made on the original manuscript are track changed to ease visibility. Furthermore, the manuscript is proofread by a native English speaker and substantial changes are made.

---

## [Decision Letter · Decision Letter 1]

Exploring the Challenges and Opportunities of Multisectoral Nutrition Programme in Ethiopia: A Qualitative Study on Combating Undernutrition during Pregnancy

PONE-D-24-40390R1

Dear Dr. Hankalo,

We’re pleased to inform you that your manuscript has been judged scientifically suitable for publication and will be formally accepted for publication once it meets all outstanding technical requirements.

Kind regards,

Dinaol Abdissa Fufa, Mph

Academic Editor

PLOS ONE

Additional Editor Comments (optional):

Reviewers' comments:

Reviewer's Responses to Questions

**Comments to the Author**

Reviewer #1: All comments have been addressed

Reviewer #3: All comments have been addressed

2. Is the manuscript technically sound, and do the data support the conclusions?

Reviewer #1: Yes

Reviewer #3: Yes

3. Has the statistical analysis been performed appropriately and rigorously?

Reviewer #1: Yes

Reviewer #3: N/A

4. Have the authors made all data underlying the findings in their manuscript fully available?

Reviewer #1: Yes

Reviewer #3: Yes

5. Is the manuscript presented in an intelligible fashion and written in standard English?

Reviewer #1: Yes

Reviewer #3: Yes

Reviewer #1: (No Response)

Reviewer #3: All previous comments have been addressed in this revision. I have nothing further to add. Excellent work.

**Do you want your identity to be public for this peer review?** For information about this choice, including consent withdrawal, please see our Privacy Policy

Reviewer #1: No

Reviewer #3: No

---

## [Editor Report · Acceptance letter]

PONE-D-24-40390R1

PLOS ONE

Dear Dr. Hankalo,

I'm pleased to inform you that your manuscript has been deemed suitable for publication in PLOS ONE. Congratulations! Your manuscript is now being handed over to our production team.

Kind regards,

on behalf of

Dr. Dinaol Abdissa Fufa

Academic Editor

PLOS ONE